# Adaptive Evolution of the Greater Horseshoe Bat *AANAT*: Insights into the Link between *AANAT* and Hibernation Rhythms

**DOI:** 10.3390/ani14101426

**Published:** 2024-05-10

**Authors:** Yanhui Zhao, Lei Wang, Sen Liu, Yingting Pu, Keping Sun, Yanhong Xiao, Jiang Feng

**Affiliations:** 1Jilin Provincial Key Laboratory of Animal Resource Conservation and Utilization, Northeast Normal University, Changchun 130117, China; zhaoyh788@nenu.edu.cn (Y.Z.); pyingt@163.com (Y.P.); fengj@nenu.edu.cn (J.F.); 2Key Laboratory of Vegetation Ecology, Ministry of Education, Changchun 130024, China; 3School of Water Conservancy & Environment Engineering, Changchun Institute of Technology, Changchun 130012, China; wanglei@ccit.edu.cn; 4College of Life Sciences, Henan Normal University, Xinxiang 453007, China; liusen2021@htu.edu.cn

**Keywords:** greater horseshoe bat, *AANAT*, evolution, climate, hibernation

## Abstract

**Simple Summary:**

Arylalkylamine N-acetyltransferase (AANAT) is an enzyme that regulates the production of melatonin, a hormone that is essential for organisms to maintain circadian and annual rhythmic behavior. The *AANAT* gene has undergone gene duplication and inactivation during evolution. Nevertheless, the majority of mammals retain a solitary copy of *AANAT* within the biological genome, and the mechanisms by which this gene responds to environmental stimuli remain poorly understood. To address this issue, we investigated the adaptive evolution of *AANAT* at both the gene and protein levels. Our study revealed the presence of multiple mutation sites within the gene encoding *AANAT*, with variation in the structure of the AANAT protein observed across different geographic populations. We identified a positive selection of *AANAT* in populations residing at higher latitudes. Furthermore, individuals exhibiting longer hibernation periods displayed significantly higher catalytic efficiency of the AANAT enzyme compared to those with minimal hibernation behavior, suggesting a potential association between *AANAT* and hibernation rhythms. This study adds to our understanding of the adaptive evolution of *AANAT* and may provide molecular evidence for hibernation rhythm adaptation in bats.

**Abstract:**

Arylalkylamine N-acetyltransferase (AANAT) is a crucial rate-limiting enzyme in the synthesis of melatonin. *AANAT* has been confirmed to be independently duplicated and inactivated in different animal taxa in order to adapt to the environment. However, the evolutionary forces associated with having a single copy of *AANAT* remain unclear. The greater horseshoe bat has a single copy of *AANAT* but exhibits different hibernation rhythms in various populations. We analyzed the adaptive evolution at the gene and protein levels of *AANAT* from three distinct genetic lineages in China: northeast (NE), central east (CE), and southwest (SW). The results revealed greater genetic diversity in the *AANAT* loci of the NE and CE lineage populations that have longer hibernation times, and there were two positive selection loci. The catalytic capacity of AANAT in the Liaoning population that underwent positive selection was significantly higher than that of the Yunnan population (*p* < 0.05). This difference may be related to the lower proportion of α helix and the variation in two interface residues. The adaptive evolution of *AANAT* was significantly correlated with climate and environment (*p* < 0.05). After controlling for geographical factors (latitude and altitude), the evolution of *AANAT* by the negative temperature factor was represented by the monthly mean temperature (r = −0.6, *p* < 0.05). The results identified the gene level variation, functional adaptation, and evolutionary driving factors of *AANAT*, provide an important foundation for further understanding the adaptive evolution of the single copy of *AANAT* in pteropods, and may offer evidence for adaptive hibernation rhythms in bats.

## 1. Introduction

Biological rhythms are crucial for the survival and reproduction of organisms. Many important physiological states and behavioral activities exhibit circadian rhythms; the seasonal reproduction and hibernation behaviors of mammals are examples of longer-period rhythms. In mammals, biorhythms are regulated by the rhythm control center that is primarily controlled by the synthesis and secretion of melatonin by the pineal gland [1,2,3]. Melatonin, as an important hormonal signal involved in the biological clock, can transmit circadian rhythm information to other structures and organs as it circulates in the blood [4,5]. The secretion cycle of melatonin is regulated by arylalkylamine N-acetyltransferase (AANAT), a crucial rate-limiting enzyme involved in the synthesis of melatonin. The normal expression of *AANAT* is essential for the periodic secretion of melatonin and the maintenance of circadian rhythms [6,7].

*AANAT* is a pleiotropic gene with evolutionary flexibility [8,9]. In invertebrates, *AANAT* acts as a detoxification agent by acetylating arylalkylamines and polyamines [10]. In vertebrates, *AANAT* regulates rhythmic behavior by producing and releasing melatonin [3]. *AANAT* functions in both circadian and annual rhythmic behavior. At night, the E-box element of the *AANAT* promoter binds to the Bmal1-Clock heterodimer of the biological clock, enhancing the transcription and expression of *AANAT* and increasing the level of melatonin [11]. The sleep duration of zebrafish (*Danio rerio*) decreased significantly compared to that of a control group after *AANAT2* gene inactivation. However, the sleep duration returned to normal after melatonin treatment [12]. In addition to circadian behavior, *AANAT* regulates annual rhythmic behavior. *AANAT* is expressed in the corpus luteum, and knocking out the melatonin receptor MT2 significantly reduced the litter size in mice [13]. The expression of *AANAT* in the brains of hibernating animals is higher than in non-hibernating animals, and exogenous melatonin treatment can prolong the hibernation time [14,15,16,17].

AANAT has been demonstrated to have adaptively evolved from early invertebrates to vertebrates. This evolution involved the loss of NV-AANAT (non-vertebrates) in agnathans and the emergence and duplication of new gene copies of VT-AANAT (vertebrates) [9]. Recent studies have found double or even multiple copies of *AANAT* in insects [18], bony fishes [19], and even-toed ungulates [20]. The loss of *AANAT* has been observed in some species of cetaceans [21], xenarthrans [22], and pangolins [23]. However, with the exception of the above-mentioned species and a few mammals, *AANAT* in most mammals exists as a single-copy gene [24] (i.e., the DNA sequence encoding the gene occurs only once within the genome). Currently, the adaptive evolutionary significance of having a single copy of *AANAT* remains unclear.

In mammals that possess one copy of *AANAT*, chiropterans serve as ecological indicator species due to their wide distribution and high level of species diversity [25,26]. Melatonin resulting from the cascading reaction regulated by the expression on *AANAT* coding sequences [27,28] plays important roles in maintaining the reproductive rhythm and hibernation behaviors of chiropterans. For example, there is a negative correlation between ovarian activity and pineal mass in the bat *Cynopterus sphinx* that may represent an adaptive strategy to cope with unfavorable seasonal conditions [29]. Melatonin also plays a role in maintaining energy balance by regulating intracellular glucose transport, a process that helps protect bats from the risk of hypoglycemia during hibernation [30] and that cooperates with insulin to stimulate leptin synthesis. This is beneficial for regulating seasonal fluctuations in body fat levels in bats, enabling them to survive challenging conditions of seasonal weather and climate [31]. Among the chiropterans, the greater horseshoe bat, *Rhinolophus ferrumequinum*, is an insectivore, widely distributed from northeast China to Yunnan, and is divided into three genetic lineages: northeast (NE), central east (CE), and southwest (SW) [32]. There is variation in hibernation rhythm among these lineages. The NE lineage, represented by the Liaoning (LN) population, exhibits a strict hibernation habit, whereas the SW lineage, represented by the Yunnan (YN) population, remains in a state of perennial activity [33]. The greater horseshoe bat is thus an ideal species for exploring the adaptive evolution of the single copy of *AANAT*.

Therefore, the present study aimed to clarify the adaptive evolution of *AANAT* at both the gene and protein levels, while identifying the forces driving the evolution of this gene in combination with environmental factors. The specific goals were to (1) assess the genetic diversity of the gene through cloning and sequencing and clarify the molecular evolutionary relationships of *AAANT* in different populations; (2) identify the selected loci and clades through selection pressure analysis; (3) evaluate whether the gene function changed during evolution; and (4) reveal the effect of ecological selection on the evolution of the gene. The integration of both gene and protein data will also contribute to understanding the molecular evidence for hibernation adaptation in greater horseshoe bats, thus potentially providing a theoretical basis for the hibernation behavior of chiropterans.

## 2. Materials and Methods

### 2.1. Sample Collection

Wing membrane tissues of greater horseshoe bats were sampled from 77 bats from 11 geographic populations from the Jilin to Yunnan provinces, including the NE lineage from the Jilin (JL) and Liaoning (LN) provinces, the CE lineage from the Beijing (BJ), Shandong (SD), Henan (HN), Anhui (AH), Zhejiang (ZJ), Shanxi (SX), and Shaanxi (SXi) provinces, and the SW lineage from the Gansu (GS) and Yunnan (YN1, YN2) provinces (Figure 1).

Mist nets were set up at dusk in the bat habitat. After identifying the species, the captured bats were sampled using a pterygium collector with a 3 mm diameter, and one or two pieces of tissue were collected from each side. The samples were stored at −80 °C until use. All field studies were conducted according to the National Wildlife Research Code.

### 2.2. DNA Amplification, Cloning, and Sequencing

Genomic DNA was extracted from the samples using an Ezup Column Animal Genomic DNA Purification Kit (Sangon Biotech, Shanghai, China), and the quality and quantity of genomic DNA were assessed using 1% agarose gel electrophoresis and a Nanodrop One ultra-micro ultraviolet spectrophotometer.

Primers were designed for chromosome segmentation using Primer v5.0 and Oligo v7.0 according to the reference sequence of the greater horseshoe bat *AANAT* gene (https://asia.ensembl.org/index.html [Accessed on 27 October 2021]). The primers 1.1FR and 1.2FR were used for the first half of *AANAT* due to a mutation site in the primer sequence, while the second half was amplified using the primer 2FR. The specific primer information is listed in Table 1. The PCR reaction system comprised 25 μL, including 2 μL of genomic DNA, 1 μL of upstream primer, 1 μL of downstream primer, 12.5 μL of Ultra HiFidelity PCR Kit II (Tiangen Biotech, Beijing, China), and 8.5 μL of ddH_2_O.

The amplification reaction conditions included initial denaturation at 94 °C for 5 min, denaturation at 94 °C for 30 s, annealing for 30 s, extension at 72 °C, and final elongation at 72 °C for 5 min (refer to Table 1 for the annealing temperatures and extension times). The quality of the PCR products was assessed at the end of PCR using 1% agarose gel electrophoresis.

DNA fragments were purified using a SanPrep Column PCR Product Purification Kit. The purified products were then transformed into *E. coli* DH5α (Sangon Biotech, Shanghai, China) and cultured in LB medium overnight. At least five individual colonies were selected for each plasmid, and PCR was performed using the M13 universal primer. Positive clones were screened and Sanger-sequenced to obtain the target sequence.

### 2.3. Recombinant Protein Preparation

In this study, AANAT recombinases were purified and expressed from the LN population that exhibits a clear hibernation rhythm and the YN population that has an irregular hibernation pattern. The protein-coding region was synthesized based on the sequencing results (Vazyme, Nanjing, China), inserted into the bacterial expression vector pGEX4T1, and verified by Sanger sequencing, enabling the expression of the AANAT protein fused with glutathione S-transferase. The protein expression strains were selected from BL21 (DE3) pLysS (Vazyme, Nanjing, China). The expression, production, and purification methods of AANAT were based on existing related studies [34].

### 2.4. Enzymatic Activity Assays

To assess whether the evolution of the greater horseshoe bat *AANAT* was accompanied by functional changes, the purified recombinant protein was utilized as an enzyme to catalyze the reaction. The detection method for AANAT enzyme activity referred to a new colorimetric method for AANAT [35]. The reaction was conducted in 96-well plates with a final volume of 100 μL per well. The components included 2 mM serotonin, 1 mM AcCoA, 2 mM DTNB, 50 mM HEPES buffer (pH 7.5), 0.1 μg AANAT, and 1 mM AcCoA. An equal volume of ddH_2_O was added to the blank control group. The experimental group (LN, YN) and the blank control group (CG) were each set up with three experimental replicates, and three groups of technical replicates were established for each experimental replicate. The reactions were monitored at room temperature using UV-vis spectroscopy at 412 nm, with readings taken every minute for 20 min. The catalytic capacity of the AANAT enzyme was measured by the amount of CoA produced per unit of time.

Due to the non-normal distribution of the data, a nonparametric Kruskal–Wallis (K–W) test was used for statistical analysis, and the Bonferroni–Holm (B–H) method was used for *p*-value correction (https://CRAN. R-project. org/package = PMCMRplus [Accessed on 21 November 2023]). An additional analysis was conducted using a generalized linear mixed model (GLMM) with the *glmer()* function of the *lme4* package that excluded the influence of technical repetitions on experimental results [36]. The response variable was enzyme reaction rate, with technical duplication as a random factor. To perform pairwise comparisons, we set CG and LN as the first groups, and the *p*-values were calculated using the *car* package.

### 2.5. Analysis of AANAT Gene Polymorphism

The SeqMan program was used to visualize the sequencing chromatogram of the sequencing data and to correct the base sequences. The MEGA v7.0 software was utilized to remove non-target base sequences from the sequencing data using the Edit Seq program. To avoid the introduction of mutations during the PCR, five clones were tested per sample to ensure the accuracy of sequencing. The genotype of each individual was determined by the measured haplotype sequence, with two base sequences if the individual was heterozygous and only one base sequence if the individual was homozygous. Different sequences obtained from different genotype data were considered as haplotypes. The genotype and single nucleotide polymorphism (SNP) analyses were performed using DnaSP v5.0 [37].

Iqtree2 was utilized to construct an unrooted evolutionary tree for greater horseshoe bats. The optimal nucleotide replacement model was calculated as K3Pu + F + R2 using the Bayesian Information Criterion and was run 1000 times. The support rates of nodes were computed using bootstrap self-help and Shimodaira–Hasegawa (SH) approximate likelihood ratios. Evolutionary tree beautification was accomplished using the online tool Interactive Tree Of Life (iTOL) (https://itol.embl.de/ [Accessed on 26 December 2023]). We also constructed an *AANAT* haplotype network using the TCS algorithm implemented in PopArt v1.7 [38] to elucidate *AANAT* gene population genetic relationships.

### 2.6. Analysis of Selective Pressure

Historical recombination events were estimated using RDP v4.0 in order to avoid false-positive selection events. Selection analysis was performed after confirming that there were no recombination sequences.

In order to detect the effects of selection on *AANAT*, the rates of synonymous and nonsynonymous substitutions at individual codons were compared separately. MEGA v7.0 was used to extract the protein-coding regions (618 bp) from the entire sequence (the *AANAT* is 3419 bp). The genotype of each individual was determined by analyzing the sequencing chromatogram obtained through Sanger sequencing. Concerning the intraspecies developmental relationships established by the neutral SNPs in previous studies [39], the guide tree of the greater horseshoe bat was drawn using the User Tree function in MEGA.

The Phylogenetic Analysis by Maximum Likelihood (PAML) codeml program was utilized for multiple model analyses [40] including the site model, the branch model, and the branch–site model. A likelihood ratio test (LRT) was performed for each model, and *p*-values were obtained. In the site model analysis, multiple built-in models were run simultaneously, including M0 (one-ratio), M1a (neutral), M2a (selection), M3 (discrete), M7 (beta), M8 (beta and ω), and M8a (beta and ω = 1). LRTs were performed on the corresponding built-in models (M0 vs. M3, M1a vs. M2a, M7 vs. M8, and M8 vs. M8a), and the posterior probability was calculated using the Bayes Empirical Bayes (BEB) method to infer positive selection sites.

In the branch model, the northeast (NE), central east (CE), southwest (SW), and northeast and central east lineages (NE + CE) were designated as foreground branches. The analysis used the built-in “one-ratio” model, the “free-ratio” model, and the “two-ratio” model. The LRT was performed on the results of all inter-model comparisons.

In the branch–site model, the setting of foreground branches was consistent with the branch model, and the method of calculating the posterior probability to infer positive selection sites was consistent with the site model.

### 2.7. Effects of Environmental Factors on AANAT Polymorphism

To investigate whether the adaptive evolution of *AANAT* has been influenced by geographical distance and climate factors, 19 bioclimatic data for 30 years for each population were downloaded from the CHELSA site based on the longitude, latitude, and altitude of sampling points recorded by GPS during field sampling (resolution: 30 arcsec, ~1 km, time period: 1981–2010). The R package “raster” [41] was used to extract 19 climate factors that included temperature and precipitation (Table 2).

Correlation analyses (Spearman statistics) of population genetic diversity and environmental factors (19 bioclimatic factors, longitude, latitude, and altitude) were implemented using the linkET package (https://github.com/Hy4m/linkET [Accessed on 2 January 2024]) mantel_test() function.

To evaluate the most critical factors affecting the genetic diversity of this gene, we first performed a collinearity analysis (SPSS v26.0) of climate factors, geographic factors, and genetic diversity, and the factors with high collinearity were removed until VIF < 2.5. The five remaining climate factors were BIO2, BIO3, BIO8, BIO13, and BIO15, and the geographic factors were altitude (Alt) and latitude (Lat). Residual haplotype diversity (Hd), average number of nucleotide differences (k), number of variable sites (S), nucleotide diversity of synonymous sites (NDSynoS), nonsynonymous sites (NonSynoS), and nucleotide diversity nonsynonymous sites (NDNonSynoS) were calculated. The association between residual factors and *AANAT* genetic diversity was calculated using Partial Mantel tests (Spearman statistics) controlling for geographic factors. The analysis used the “hmisc” package.

## 3. Results

### 3.1. AANAT Gene Characteristics

A total of 357 *AANAT* sequences were obtained after correction and combination, with 18 homozygotes and 59 heterozygotes. Among the 171 SNPs, there were 18 coding region polymorphic sites (cSNPs), including four nonsynonymous mutation sites. The population fixation index (*F_ST_*) was 0.35. Neutrality tests yielded negative results under both Fu and Li’s D and Fu and Li’s F models (*p* < 0.05).

Populations at higher latitudes exhibit greater *AANAT* genetic diversity, as indicated by higher values of the average nucleotide diversity (π), an average number of nucleotide differences (K), θ_per sequence_ from the total number of mutations (θ-W_seq_), and θ_per site_ from the total number of mutations (θ-W_site_). This suggests that genetic diversity is greater in populations at higher latitudes (LN and HN), while populations at lower latitudes (YN) show less genetic diversity. There was little difference between populations (Figure 2a). In various lineages, the values of π, K, and polymorphic sites (S) were highest in the CE lineage, followed by NE, and lowest in SW. θ-W_seq_ and θ-W_site_ exhibited the highest values in NE, followed by CE, and the lowest in SW (Figure 2b).

### 3.2. AANAT Phylogeny

The phylogenetic tree and haplotype network diagrams constructed from the *AANAT* sequences showed that the SW lineage of greater horseshoe bats (GS, YN) clustered separately, and the NE lineage clustered with the CE lineage (Figure 3a,b). The haplotype network diagram showed that most individuals in Yunnan shared the same haplotype, and the large number of haplotypes found in different populations may be the result of the creation of new genes during an ancient gene duplication event, and the subsequent evolution of these genes individually [42].

### 3.3. Selective Pressure on AANAT

Two identical positive selection sites were detected in M2a and M8 at 70 (I 0.693 V) and 187 (A 0.906 T), respectively (Table 3).

In the branch model, the NE, CE, NE + CE, and SW lineages were identified as foreground branches, and the LRT *p*-values for the “one-ratio” model compared to the “two-ratio” model (two-ratio vs. one-ratio) were not significant (*p* > 0.5). Compared with the alternative hypothesis in the “double ratio” model and the corresponding null model (two-ratio vs. two-ratio, ω_1_ = 1), the LRT test was significant (*p* < 0.001) when the NE, CE, and NE + CE lineages were used as foreground branches (Table 4). The results showed that the NE and CE lineages had a rapid evolutionary rate.

In the branch–site model, when the NE + CE lineage was taken as the foreground branch, although the *p*-value of the likelihood ratio test was not significant (*p* > 0.1), two positive selection sites were detected, which were consistent with the positive selection sites detected in the site model. These were at 70 (I 0.587 V) and 187 (A 0.811 T), respectively (Table 5), indicating that the NE and CE lineages had been subjected to positive selection pressure.

### 3.4. Comparison of the Catalytic Capacity of AANAT in Various Geographic Populations

The CoA production within 20 min was in the order LN > YN > blank control group (CG) (Figure 4a). There were significant differences between the three treatment groups, among which the catalytic capacity of AANAT recombinase in the LN and YN populations was significantly higher than that in the CG (Z = −6.7, *p_adj_* < 0.0001; Z = −4.7, *p_adj_* = 0.01). Additionally, the catalytic capacity of AANAT recombinase in the LN population was significantly higher than that in the YN population (Z = 2.2, *p_adj_* = 0.02) (Figure 4b).

GLMM analysis showed that the CG was significantly different from the LN and YN populations (*t* = 36.45, *p* < 0.001; *t* = 23.49, *p* < 0.001), and there were also significant differences between LN and YN (*t* = −12.28, *p* < 0.001) (Table 6).

### 3.5. Prediction of the AANAT Protein Structure

The results of the secondary structure prediction revealed that the proportion of α helix in the LN population was 20%, while in the YN population, it was 21%. The main distinction was the difference in the length of the α helix, about 170 amino acids (Figure 5a). The results of tertiary structure prediction showed that the spatial folding of the LN population differed from that of the YN population, with an RMSD value of 0.203. Protein characterization predicted that leucine at position 32 of the YN population was located on the protein surface, while proline at position 33 of the LN population was also located on the protein surface (Figure 5b).

### 3.6. Drivers of Adaptive Evolution in AANAT

The genetic diversity of *AANAT* in different geographic populations was significantly correlated with climate factors related to temperature and precipitation (Figure 6). π and K were significantly correlated with BIO4 (*p* < 0.05). S was significantly correlated with BIO14, BIO17, and BIO19 (*p* < 0.05). Twelve climate-related characteristics showed significant correlations with *AANAT* genetic diversity. The genetic diversity of *AANAT* was significantly correlated with the geographic factors of latitude (Lat) and altitude (Alt). This suggests that environmental pressures, such as temperature, humidity, and changes in geographical scale have impacted the adaptive evolution of *AANAT*. The crucial factors were further examined using Partial Mantel analysis.

BIO2 showed a significant negative correlation with nonsynonymous sites, independent of other variables (r = −0.6, *p* < 0.05) (Figure 7a). This correlation remained significant even after accounting for the geographic factors of latitude and altitude (r = −0.6, *p* < 0.05). BIO3 and BIO13 were significantly correlated with the vS without considering other variables (r = −0.7, *p* < 0.05; r = 0.6, *p* < 0.05) (Figure 7b,d), but the correlation was not significant after controlling for geographic factors (*p* > 0.05). BIO8 was significantly correlated with altitude (r = 0.8, *p* ≤ 0.001) (Figure 7c), and BIO15 was not significantly correlated with any climatic or geographic factors (*p* > 0.05) (Figure 7e).

## 4. Discussion

The sequences of *AANAT* were examined, and four out of 171 SNPs were nonsynonymous mutations. The analysis indicated that genetic differentiation occurred in *AANAT* and did not follow a neutral model, as well as that signals of selection were present. The mismatch distribution curve of *AANAT* showed a bimodal distribution, indicating that the population size remained stable and did not experience expansion or continuous growth, suggesting that departure from expectations under neutrality was not caused by population expansion, but may have been influenced by selection. Studies have shown that the genetic variation of populations, as indicated by neutral markers, is consistent with the isolation-by-distance (IBD) model. This means that genetic variation gradually increases with geographic distance [43], suggesting the role of genetic drift. However, our neutrality test revealed that the gene sequences did not adhere to the neutral model of evolution, instead exhibiting signals of selective pressure.

*AANAT* is highly conserved in vertebrates, and it may have evolved early on through horizontal gene transfer [44] and thereafter was driven by natural selection [24]. The genetic diversity of the NE and CE lineages with longer hibernation times was greater than that of the SW lineage, indicating evolutionary potential and the ability to adapt to environmental changes [45]. Compared to the phylogenetic tree constructed using neutral markers [32], the *AANAT* phylogenetic tree revealed a mixed clustering of NE and CE lineages. This may be associated with the unique functions related to *AANAT*. The NE and CE lineages have a longer hibernation period, while the YN population of the SW lineage exhibits almost no hibernation behavior. In particular, two positively selected sites (70, I 0.693 V, 187, A 0.906 T) were detected in the NE and CE lineages, indicating the adaptive evolution of this gene. *AANAT* appears to be subject to specific selection pressures in different groups. For example, the functional diversification of the insect *AANAT* family has been shaped by positive selection [46]. The whole-gene replication process of *AANAT* in bony fishes may be attributed to its rapid evolution that facilitates genetic changes in response to environmental stress [47]. In a study of human delayed sleep phase syndrome (DSPS) focusing on *AANAT*, position 129 was identified as an SNP. The mutation of alanine to threonine at this site may be associated with differences in human sleep patterns [48]. This mutation was also identified in the present study and showed signals of positive selection, suggesting a potential relationship with differences in the timing of hibernation in the greater horseshoe bat.

The activity of *AANAT* is closely and positively correlated with the production of melatonin [49]. Differences in the catalytic capacity of AANAT will affect variation in the secretion of melatonin. In this study, we found that the expression of *AANAT* was increased following positive selection for a longer hibernation period. By regulating the secretion of melatonin, *AANAT* provides hibernation signals for hibernating organisms [50,51]. It also regulates neuroprotection during hibernation [52,53], and energy homeostasis [54]. The endocrine signals generated by the interaction of melatonin and photoperiod also impact the synthesis of leptin and growth hormone, both of which are related to the regulation of body weight [51] and work together to achieve the hypothermia required for hibernation, inhibiting metabolism and the endocrine response to fasting. A study of Atlantic salmon (*Salmo salar*) and European seabass (*Dicentrarchus labrax*) found that variation in melatonin production may be associated with differences in their respective marine habitats [55]. Reduced melatonin secretion weakens rhythmic behaviors in organisms, such as unihemispheric slow-wave sleep and long-term alertness behavior in cetaceans [56]. Duplication of the reindeer *AANAT* may have occurred in order to maintain active circadian and annual rhythms in polar regions where there is a lack of a consistent photoperiod [57,58]. By linking the physiological tolerance of organisms to evolutionary and biogeographic processes, researchers have found that *AANAT* duplication and loss occurred in the middle Eocene when a more seasonal climate emerged [24,59]. Based on the potential causal relationship between genetic and phenotypic adaptation [60], we hypothesized that the adaptive evolution of *AANAT* may be linked to its adjustment of hibernation rhythms.

It has been confirmed that the gene duplication and inactivation of *AANAT* were driven by environmental selection pressure [24]. *AANAT* exists in three copies (*AANAT1a*, *AANAT1b*, and *AANAT2*) in bony fishes [61]. The dynamical differences and different temperature–activity relationships exhibited by *AANAT1a* and *AANAT1b* may represent evolutionary responses to different environmental stresses [62,63]. The loss of *AANAT* in bottle-nosed dolphins [64] may be related to environmental factors such as a nonconstant photoperiod and limited temperature fluctuations in the ocean. The function of *AANAT* has evolved from a primitive detoxification role to promoting melatonin synthesis, thereby aiding in the adaptation of the eye and retina to the photoperiod. This significant change in function is believed to be the outcome of sustained stress from environmental and metabolic factors [9,65]. Furthermore, research has demonstrated that the expression of genes associated with the circadian rhythms of *Daphnia magna* exhibits a geographical gradient, indicating local adaptation to environmental stress [66]. Temperature may have been a selective pressure influencing the evolution of genes associated with circadian systems [67]. Obligatory diurnal and nocturnal mammals are more than twice as likely to adapt to climate change as those with flexible schedules. The response of mammals to climate change has been primarily influenced by their size and the timing of their activities [68]. In this study, we found that the variation in *AANAT* was primarily influenced by negative temperature factors represented by BIO2. This suggests that the genetic variation in species with a single copy of *AANAT* has been influenced by environmental selection pressure, similar to other functional genes such as the MHC genes that are significantly correlated with environmental temperature and humidity [69].

## 5. Conclusions

In this study, we cloned, sequenced, and analyzed the evolution of *AANAT* at the gene and protein levels in 11 geographic populations of greater horseshoe bats in China. We discovered that populations with longer hibernation periods exhibited greater levels of genetic diversity and were subject to positive selection. In vitro enzyme activity validation experiments demonstrated that the catalytic capacity of the AANAT enzyme was higher in populations undergoing positive selection. Therefore, we suggest that the variation in *AANAT* may be related to its role in the adaptation of hibernation rhythms and is negatively influenced by temperature factors. This study contributes to a better understanding of the evolution of *AANAT* and provides a theoretical basis for investigating the adaptive significance of a single copy of *AANAT* in chiropterans. Additionally, our results may provide evidence for the adaptation of the hibernation rhythm of greater horseshoe bats.

## Figures and Tables

**Figure 1 animals-14-01426-f001:**
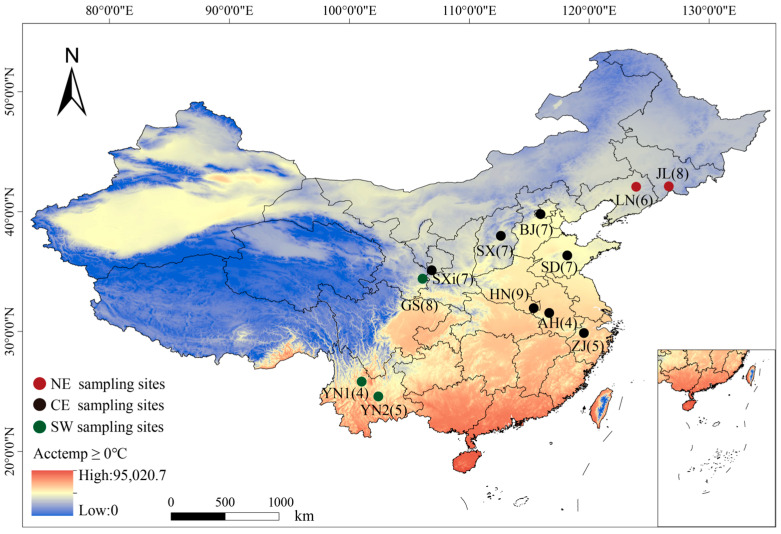
Sampling distribution map of *Rhinolophus ferrumequinum*. The numbers in parentheses are the number of samples. The bottom left color diagram explains the colors in the accumulated temperature heat map. Acctemp = accumulated temperature. Different populations belong to different genetic lineages, where red represents the northeast (NE), black indicates the central east (CE), and green indicates the southwest (SW).

**Figure 2 animals-14-01426-f002:**
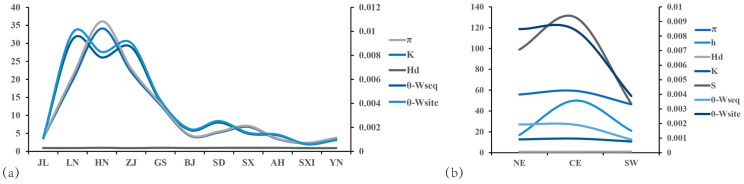
*AANAT* genetic diversity parameters of various (**a**) populations and (**b**) lineages. π, θ-W_seq_, and θ-W_site_ refer to the secondary coordinates, and the remaining parameters refer to the primary coordinates. π = average nucleotide diversity; K = average number of nucleotide differences; S = number of polymorphic sites; h = number of haplotypes; Hd = haplotype diversity; θ-W_seq_ = θ_per sequence_ from the total number of mutations; θ-W_site_ = θ_per site_ from the total number of mutations. The population and lineage abbreviations are shown in Figure 1.

**Figure 3 animals-14-01426-f003:**
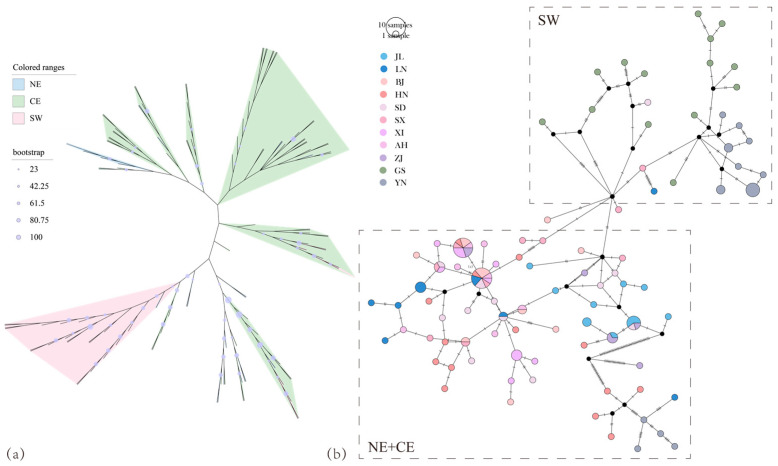
Molecular evolutionary relationships of the *AANAT*. (**a**) Unrooted evolutionary tree for *AANAT* based on the maximum likelihood method. The purple circles of different sizes indicate bootstrap values for the evolutionary trees. (**b**) *AANAT* nucleotide level haplotype network diagram. The size of the circle indicates the size of the sample. Nodes are proportional to the number of bats carrying each haplotype and are colored according to the bat population (see the illustration).

**Figure 4 animals-14-01426-f004:**
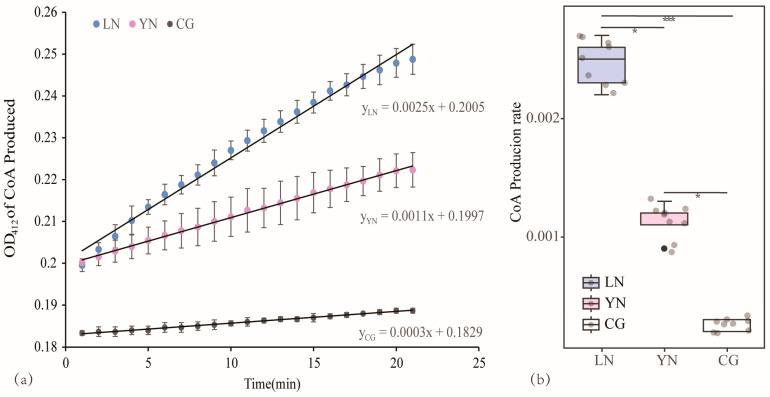
Comparison of catalytic efficiency of AANAT recombinase in different geographical populations. (**a**) Time curve of CoA production between the experimental group (YN, LN) and the blank group (CG). (**b**) Comparison of catalytic capacities of AANAT recombinase in different geographical populations. YN = YN population, LN = LN population, CG = blank control group. The * symbol means 0.01 < *p* < 0.05, and the *** symbol means *p* ≤ 0.001.

**Figure 5 animals-14-01426-f005:**
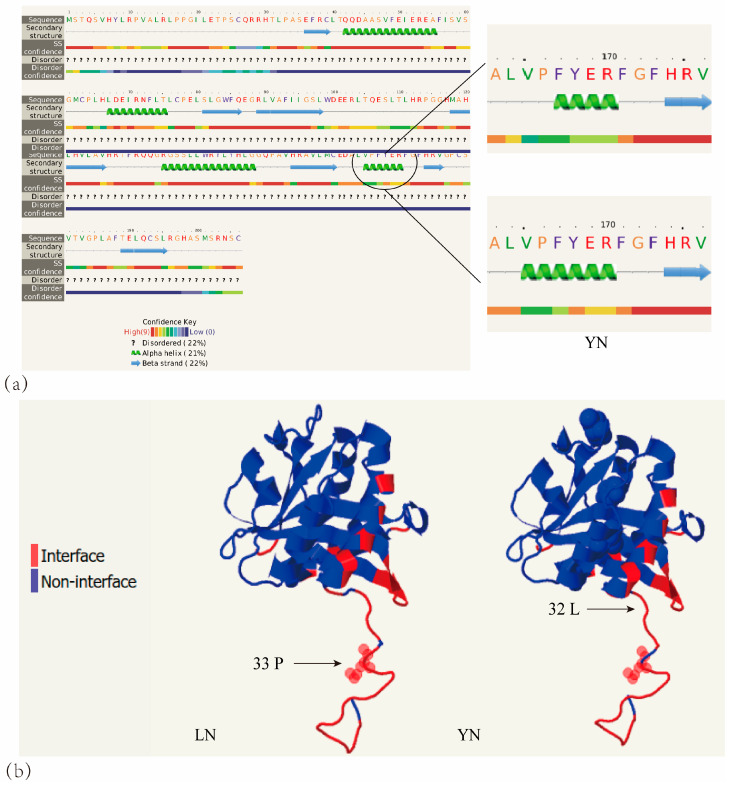
AANAT protein prediction results. (**a**) Areas of the secondary structure of the AANAT protein with differences between LN and YN. The green helix in the diagram indicates an alpha helix and the blue arrow indicates a beta fold. (**b**) Prediction results of AANAT protein interface characterization in YN and LN. The red color in the diagram indicates residues located on the surface of the protein and the blue color indicates residues located inside the protein.

**Figure 6 animals-14-01426-f006:**
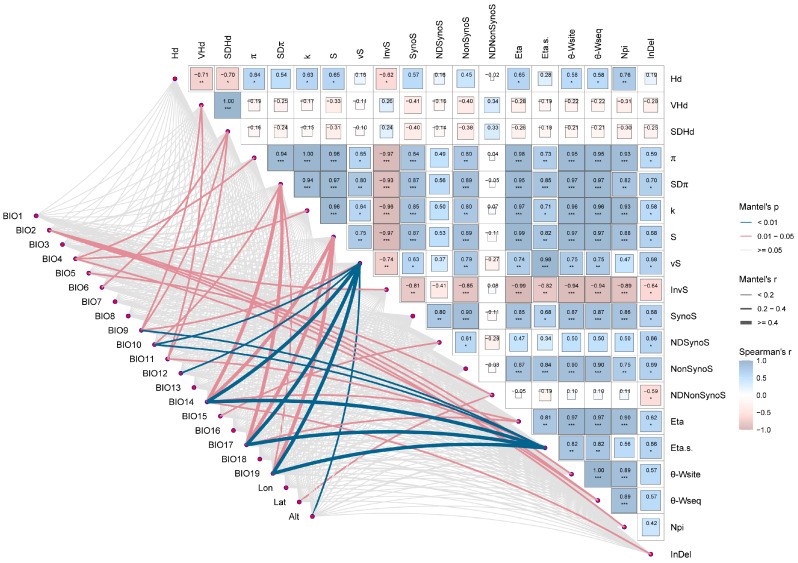
Correlations between genetic diversity and climate factors. The color gradient and edge width in the heat map represent Spearman correlation coefficients. The pink line indicates that there is a significant correlation between *AANAT* genetic diversity and climate factors, and the blue line indicates that there is a significant correlation between *AANAT* genetic diversity and climate factors. Hd = haplotype diversity; VHd = variance of haplotype diversity; SDHd = standard deviation of haplotype diversity; π = average nucleotide diversity; SDπ = standard deviation of π; k = average number of nucleotide differences; S = number of polymorphic sites; vS = singleton variable sites; InvS = invariable sites; SynoS = synonymous sites; NDSynoS = nucleotide diversity synonymous sites; NonSynoS = nonsynonymous sites; NDNonSynoS = nucleotide diversity nonsynonymous sites; Eta = total number of mutations; Eta.s. = total number of singleton mutations; θ-W_site_ = θ_per site_ from Eta; θ-W_seq_ = θ_per sequence_ from Eta; Npi = number of parsimony informative sites; InDel = indel diversity. The * symbol means 0.01 < *p* < 0.05, the ** symbol means 0.001 < *p* ≤ 0.01, and the *** symbol means *p* ≤ 0.001.

**Figure 7 animals-14-01426-f007:**
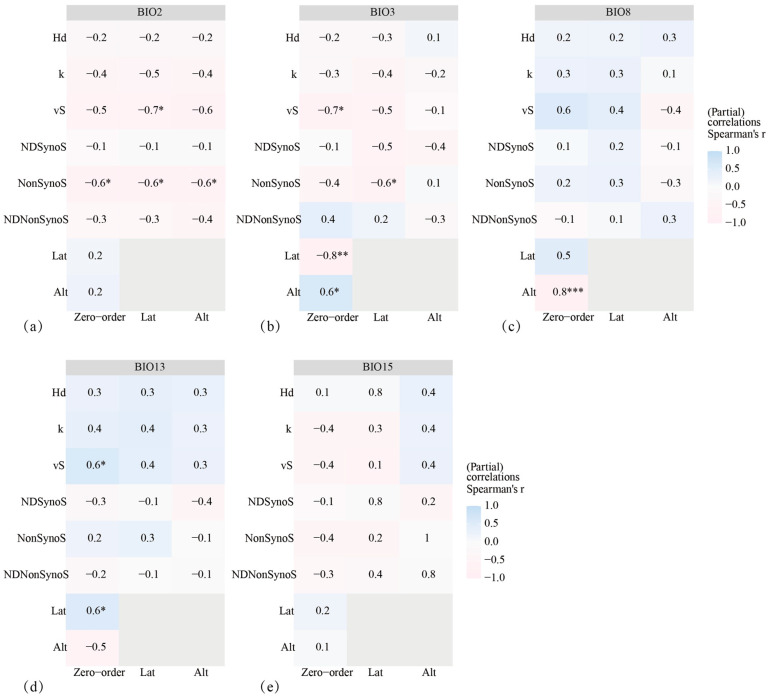
Partial Mantel test for climatic factors and genetic diversity. (**a**) Monthly mean temperature (BIO2); (**b**) isothermality (Bio2/Bio7) (×100) (BIO3); (**c**) mean temperature of wettest quarter (BIO8); (**d**) precipitation of the wettest month (BIO13); and (**e**) precipitation seasonality (coefficient of variation) (BIO15). The horizontal coordinate represents the control variable. Pink-colored blocks in the heatmap indicate a negative correlation between *AANAT* genetic diversity and climate factors, and blue-colored blocks indicate a positive correlation. * 0.01 < *p* < 0.05; ** 0.001 < *p* ≤ 0.01; *** *p* ≤ 0.001. Hd = haplotype diversity; k = average number of nucleotide differences; S = number of polymorphic sites; NDSynoS = nucleotide diversity synonymous sites; NonSynoS = nonsynonymous sites; NDNonSynoS = nucleotide diversity nonsynonymous sites; Lat = latitude; Alt = altitude.

**Table 1 animals-14-01426-t001:** Primer design and PCR information.

Name	Base Sequence (5′-3′)	Annealing	Extending	Products
1.1F	TCAACACCTAGCAAGAGC	55 °C	2 min	1981 bp
1.1R	GTCCCAAAGTGAACCGAT			
1.2F	CTGTTACCTGCGGCTCAAC	57 °C	2.5 min	2394 bp
1.2R	TGTCACCTCTGCGGATACCT			
2F	TGCCGCAGCCTTCATCTCTGTCTCG	57 °C	1.8 min	1712 bp
2R	CCTCAGGAAGAATGAAAGCTGGAACCTT			

**Table 2 animals-14-01426-t002:** Abbreviations of 19 bioclimatic factors downloaded from CHELSA.

Abbreviation	Bioclimatic Variable
BIO1	Annual Mean Temperature
BIO2	Monthly Mean Temperature
BIO3	Isothermality (Bio2/Bio7) (×100)
BIO4	Temperature Seasonality (standard deviation) (×100)
BIO5	Max Temperature of Warmest Month
BIO6	Min Temperature of Coldest Month
BIO7	Temperature Annual Range (Bio5-Bio6)
BIO8	Mean Temperature of Wettest Quarter
BIO9	Mean Temperature of Driest Quarter
BIO10	Mean Temperature of Warmest Quarter
BIO11	Mean Temperature of Coldest Quarter
BIO12	Annual Precipitation
BIO13	Precipitation of Wettest Month
BIO14	Precipitation of Driest Month
BIO15	Precipitation Seasonality (Coefficient of Variation)
BIO16	Precipitation of Wettest Quarter
BIO17	Precipitation of Driest Quarter
BIO18	Precipitation of Warmest Quarter
BIO19	Precipitation of Coldest Quarter

**Table 3 animals-14-01426-t003:** The positive selection analyses for the site model.

Model	Parameters Estimate	Ln L	df	2ΔLn L		*p*-Value	Positively Selected Sites
M0 (one-ratio)	ω = 0.05305	−1010.615442			29		None
M1a (neutral)	p_0_ = 0.95691, p_1_ = 0.04309 ω_0_ = 0.00000, ω_1_ = 1.00000	−1002.789947			30		None
M2a (selection)	p_0_ = 0.97461, p_1_ = 0.00000, p_2_ = 0.02539 ω_0_ = 0.00490, ω_1_ = 1.00000, ω_2_ = 1.93757	−1002.481582			32		70I (0.573), 187A (0.800)
M3 (discrete)	p_0_ = 0.29208, p_1_ = 0.68253, p_2_ = 0.02539 ω_0_ = 0.00490, ω_1_ = 0.00490, ω_2_ = 1.93760	−1002.481582			33		None
M7 (beta)	*p* = 0.01000, q = 0.17324	−1004.096785			30		None
M8 (beta and ω > 1)	p_0_ = 0.97460, *p* = 0.51911, q = 99.00000 (p_1_ = 0.02540), ω = 1.93628	−1002.482059			32		70I (0.693), 187A (0.906)
M8a (beta and ω = 1)	p_0_ = 0.95691, *p* = 0.00500, q = 2.54319 (p_1_ = 0.04309), ω = 1.00000	−1002.789924			31		None
LRT of variable ω values among branches							
M0 vs. M3			4	16.26772		0.05 < *p* < 0.1	
M1a vs. M2a			2	0.61673		0.5 < *p* < 0.9	
M7 vs. M8			2	3.229452		0.1 < *p* < 0.5	
M8 vs. M8a			1	0.61573		0.1 < *p* < 0.5	

Significant value of *p* < 0.05 for LRT.

**Table 4 animals-14-01426-t004:** The positive selection analysis for the branch model.

Model	Foreground Branch	Parameter Estimates	Ln L	df	2ΔLn L		*p*-Value
One-ratio		ω = 0.05304	−1010.615442			30	
Free-ratio			−1009.141884			57	
Two-ratio	NE	ω_0_ = 0.05688, ω_1_ = 0.04539	−1010.584015			31	
	CE	ω_0_ = 0.04230, ω_1_ = 0.05905	−1010.545700			31	
	SW	ω_0_ = 0.05438, ω_1_ = 0.00010	−1010.459894			31	
	NE + CE	ω_0_ = 0.00010, ω_1_ = 0.05438	−1010.459894			31	
Two-ratio, ω = 1	NE	ω_0_ = 0.05686, ω_1_ = 1.00000	−1024.101938			30	
	CE	ω_0_ = 0.04229, ω_1_ = 1.00000	−1034.724656			30	
	SW	ω_0_ = 0.05305, ω_1_ = 1.00000	−1010.615980			30	
	NE + CE	ω_0_ = 0.00010, ω_1_ = 1.00000	−1048.017833			30	
LRT of variable ω values among branches							
Free-ratio vs. one-ratio				27	2.947116		1
LRT of ω at specific lineages							
Two-ratio vs. one-ratio	NE			1	0.062854		0.5 < *p* < 0.9
	CE			1	0.139484		0.5 < *p* < 0.9
	SW			1	0.311096		0.5 < *p* < 0.9
	NE + CE			1	0.311096		0.5 < *p* < 0.9
Two-ratio vs. two-ratio (ω_1_ = 1)	NE			1	27.035846		*p* < 0.001
	CE			1	48.357912		*p* < 0.001
	SW			1	0.312172		0.5 < *p* < 0.9
	NE + CE			1	75.115878		*p* < 0.001

The branches that were defined as the foreground branches are shown in the Section 2. Significant values of *p* < 0.05 for LRT.

**Table 5 animals-14-01426-t005:** The positive selection analyses for the branch–site model.

Foreground Branch	Parameter Estimates	Ln L	df	2ΔLn L	np	*p*-Value	Selected Sites
NE	Null	−1001.926974	1	0.134988	32	0.9 < *p* < 0.97	None
	*p*_0_ = 0.96733, *p*_1_ = 0.03267, *p*_2a_ = 0.00000, *p*_2b_ = 0.00000						
	Background: ω_0_ = 0.00000, ω_1_ = 1.00000, ω_2a_ = 0.00000, ω_2b_ = 1.00000						
	Foreground: ω_0_ = 0.00000, ω_1_ = 1.00000, ω_2a_ = 1.00000, ω_2b_ = 1.00000						
	Aternative	−1001.859480			33		
	*p*_0_ = 0.96160, *p*_1_ = 0.02817, *p*_2a_ = 0.00994, *p*_2b_ = 0.00029						
	Background: ω_0_ = 0.00000, ω_1_ = 1.00000, ω_2a_ = 0.00000, ω_2b_ = 1.00000						
	Foreground: ω_0_ = 0.00000, ω_1_ = 1.00000, ω_2a_ = 1.00000, ω_2b_ = 1.00000						
CE	Null	−1001.926974	1	0	32	1	None
	*p*_0_ = 0.96733, *p*_1_ = 0.03267, *p*_2a_ = 0.00000, *p*_2b_ = 0.00000						
	Background: ω_0_ = 0.00000, ω_1_ = 1.00000, ω_2a_ = 0.00000, ω_2b_ = 1.00000						
	Foreground: ω_0_ = 0.00000, ω_1_ = 1.00000, ω_2a_ = 1.00000, ω_2b_ = 1.00000						
	Aternative	−1001.926974			33		
	*p*_0_ = 0.96733, *p*_1_ = 0.03267, *p*_2a_ = 0.00000, *p*_2b_ = 0.00000						
	Background: ω_0_ = 0.00000, ω_1_ = 1.00000, ω_2a_ = 0.00000, ω_2b_ = 1.00000						
	Foreground: ω_0_ = 0.00000, ω_1_ = 1.00000, ω_2a_ = 1.00000, ω_2b_ = 1.00000						
NE + CE	Null	−1002.650389	1	0.666436	32	0.1 < *p* < 0.5	70I (0.587) 187A (0.811)
	*p*_0_ = 0.95622, *p*_1_ = 0.00000, *p*_2a_ = 0.04378, *p*_2b_ = 0.00000						
	Background: ω_0_ = 0.00000, ω_1_ = 1.00000, ω_2a_ = 0.00000, ω_2b_ = 1.00000						
	Foreground: ω_0_ = 0.00000, ω_1_ = 1.00000, ω_2a_ = 1.00000, ω_2b_ = 1.00000						
	Aternative	−1002.317171			33		
	*p*_0_ = 0.97334, *p*_1_ = 0.00000, *p*_2a_ = 0.02666, *p*_2b_ = 0.00000						
	Background: ω_0_ = 0.00396, ω_1_ = 1.00000, ω_2a_ = 0.00396, ω_2b_ = 1.00000						
	Foreground: ω_0_ = 0.00396, ω_1_ = 1.00000, ω_2a_ = 1.93612, ω_2b_ = 1.93612						
SW	Null	−1002.789947	1	0	32	1	None
	*p*_0_ = 0.95691, *p*_1_ = 0.04309, *p*_2a_ = 0.00000, *p*_2b_ = 0.00000						
	Background: ω_0_ = 0.00000, ω_1_ = 1.00000, ω_2a_ = 0.00000, ω_2b_ = 1.00000						
	Foreground: ω_0_ = 0.00000, ω_1_ = 1.00000, ω_2a_ = 1.00000, ω_2b_ = 1.00000						
	Aternative	−1002.789947			33		
	*p*_0_ = 0.95691, *p*_1_ = 0.04309, *p*_2a_ = 0.00000, *p*_2b_ = 0.00000						
	Background: ω_0_ = 0.00000, ω_1_ = 1.00000, ω_2a_ = 0.00000, ω_2b_ = 1.00000						
	Foreground: ω_0_ = 0.00000, ω_1_ = 1.00000, ω_2a_ = 1.00000, ω_2b_ = 1.00000						

**Table 6 animals-14-01426-t006:** The results of the estimation of the effect of the group in the GLMM on the catalytic rate of the AANAT enzyme.

Fixed Effects	Estimate	Std. Error	*t*-Value	*p*-Value
Intercept 1	−8.23400	0.05266	−156.35	<0.001 ***
LN vs. CG	2.23374	0.06129	36.45	<0.001 ***
YN vs. CG	1.44237	0.06141	23.49	<0.001 ***
Intercept 2	−6.00039	0.04563	−131.51	<0.001 ***
YN vs. LN	−0.79205	0.06452	−12.28	<0.001 ***
CG vs. LN	−2.22912	0.06452	−34.55	<0.001 ***

LN = LN population, YN = YN population, CG = blank control group. The *** symbol indicates *p* ≤ 0.001.

## Data Availability

*AANAT* sequencing data have been deposited in the NCBI (PP358600–PP358728).

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
