# Peer review of "Adaptive Evolution of the Greater Horseshoe Bat AANAT: Insights into the Link between AANAT and Hibernation Rhythms"

_animals, 2024, doi:10.3390/ani14101426_

Round 1
Reviewer 1 Report
Comments and Suggestions for Authors
Review Manuscript ID animals-2921877
Title :
Adaptive evolution of the greater horseshoe bat AANAT : insights into the link between AANAT and hibernation rhythms
In the present manuscipt, the authors first summarize the extent state of knowledge on the role of AANAT gene in regulating circadian and seasonal rhythms in a wide range of species, and follow by a concise presentation of the study species, the Greater horseshoe bat Rinolophus ferrumequinum, its distribution in China and the occurrence of three lineages North-East, Central-East and South-West provinces of the country. The authors describe sample collection and DNA extraction and amplification, recombinant protein preparation and analysis of AANAT gene polymorphism in relation to environmental variables. Results convincingly show that sequence varaition associates with in vitro differences in enzymatic activity of the AANAT. The authors then discuss implications of their finding for the better understanding of hibernation periods in Greater horseshoe bat, and potentially other organisms.
General comments:
Introduction: The last paragraph (p.2, l.88–96) summarises the results of the study and should be rephrased to describe the aims of the study, working hypothesis and expected outcomes.
Material and methods: Using appropriate terminology greatly ease the reading and understanding (ie, electropherogram rather than peak plots, …).
Results: a) Define symbols, acronyms or jargon when first used in a section, b) Results section should be limited to the mere description of results. Please make sure to purge the text from interpretation and discussion.
Discussion: Evidence is missing to argue that sequence variation at the AANAT induces changes in hibernation period in Greater horseshoe bat, and the paper would benefit from tuning down such statements (ie, avoid suspicion of overselling results).
Specific comments:
p.2, l.71: Add a definition of biodiversity and to which levels does this term refers to (diversity at species level, genetic diversity, phenotypic diversity, all levels?).
p.2, l.72: “AANAT and its expression product, melatonin…“. Melatonin is not produced by the AANAT gene. Is melatonin the only molecule resulting from the cascading reaction regulated by the expression on AANAT coding sequences? This remark also holds for the interpretation of the results. Regulation of melatonin production by AANAT is one, out of potentially several, factors affecting hibernation in the species.
p.2, l.80–81: Remove “the“ and add the species name in English: “Among the chiropterans, Greater horseshoe bat, Rhinolophus ferrumequinum, is widely distributed…“. Species name is written with capital G (Greater). Correct throughout the manuscript.
p.3, l.102–106:
a) Text and Figure 1 may contain an error, as two populations are labelled Shanxi in the text, but were apparently located in different provinces according to borders drawn on the background map.
b) Figure 1 illustrate the distribution of the lineages and would benefit from the integration of some of the environmental variables described, for instance as heatmaps of precipitations or monthly temperature.
p.3, l.123–126: PCR reaction starts with an initial denaturation (not a “pre-denaturation“ step) and ends up after a final elongation step (not a “re-extension“ step).
p.3, l.132: Describe the method used for sequencing (Sanger or NGS)?
p.4, l.142: Sub-sections 2.3 and 2.4 both refer to “Recombinant protein preparation“.
p.4, l.156–158: Statistical test is fine. I just wonder why using a nonparametric test? Mixed-effect model, would account for pseudo-replication (technical replicates are not independent) and provide a quantitative estimate of effect size in the three treatments.
p.4, l.160: Use accepted terminology “sequencing chromatogram” or “electropherogram”, rather than the cryptic “peak plot“.
p.4, l.162: “The haplotype of each individual was determined…“ AANAT gene is located on autosomes, hence, individuals carry two copies of the gene (genotype rather than haplotype).
p.4, l.164: “…, where mutation site introduced by PCR amplification was excluded“. Did the “Ultra HiFidelity“ polymerase used in the PCR reaction introduced mutations, or was the mutation site within primer sequences?
p.4, l.169–174: Three sentences in a row, combining jargon and acronyms, make it difficult to read and understand. Needs extensive rephrasing.
p.5, l.176–177: Why removing recombinant sequences?
p.5, l.178–183: I’m not sure if I got it right.
a) Suggestion: MEGA v7.0 was used to extract the protein-coding regions (618 bp) from the entire sequence (6087 bp according to Table 1).
b) How did phase haplotypes to determine individual genotypes?
c) Missing a brief description of the method and tools used to infer intraspecific phylogeny of Greater horseshoe bat in China (Zhao 2020, ref n°36).
p.5, l.184–191: Define symbols, acronyms or jargon when first used in the text.
p.5, l.206–p.6, l.220: Were correlation analyses conducted on the entire set of 19 variables (as described in the Results section), or on the five climate factors remaining after filtering for collinearity (as described in the Material & method section)?
p.6, l.223–224: I didn’t get the procedure followed by the authors to convert haploid sequences obtained after cloning into diploid genotypes?
p.6, l.228: “…did not follow a neutral model and that a signal of selection is was present.“ Interpretation of data (not to be included in the Results section). By the way, departure from expectations under neutrality may result from processes other than selection (demography).
p.6, l. 230: Define symbols, acronyms or jargon when first used in a section.
p.7, l. 260: Part of the sentence is duplicated.
Tables 3, 4 and 5: Missing a detailed description of table contents in the legend. Format tables (missing separators between categories).
p.13, l.300–p.15, l.332:
a) How was nucleotide diversity (π) estimated? Nucleotide diversity (π) is usually quantified as the proportion of nucleotide differences between sequences (= K). In that case, it would not be surprising that the two variables are correlated (r = 1, p < 0.001).
b) According to Material & method section, only variables BIO2, BIO3, BIO8, BIO13 and BIO15, were retained and all other environmental variables were excluded (collinearity threshold). Results section describes correlation between genetic variables (some of them correlated) and a set of environmental factors (BIO4, BIO14, BIO17 and BIO19). Why did the authors use them in the analysis?
p.15, l.336: First occurrence of the estimation of population genetic differentiation (FST = 0.35). Add into Results section.
p.15, l.342–346: Please also discuss other phenomena that may lead to reject the hypothesis of neutrality (population expansion, …).
p.16, l.363–365: The authors convincingly showed in vitro difference in activity between recombinase resulting from the expression of coding regions sequences retrieved in YN and LN. As acknowledged by the authors AANAT is a pleiotropic gene, and confirmation that differences in gene sequences affected hibernation period in Greater horseshoe bat are missing.
Recommendations
The authors convincingly showed in vitro difference in activity between genetic variants (alleles) found in different location of the Greater horseshoe bat distribution range in China.
The study is not hampered by major flaw, was mostly well written. I don’t have any argument to reject the paper. Nonetheless, major changes are required before the manuscript can be accepted for publication.
Comments on the Quality of English Language
The quality of English language is generally good. However, several sentences throughout the manuscript are too complex and difficult for the reader to understand. In particular, make sure to use accepted terminology.
Author Response
We have uploaded the ”Cover Letter to Reviewer1“ to the attachment, please check.

Reviewer 2 Report
Comments and Suggestions for Authors
REVIEW
Bats are important ecological and evolutionary factors in the bat region. They are known to have diverse dietary habits and play a key role in reforestation (pollinators and seed dispersers) and pest control. Population trends of the large horseshoe in Eastern Europe including Poland were analyzed.
Bats are particularly susceptible to environmental changes due to their low reproductive rate, longevity and high metabolic rate. They respond to environmental changes, from changes in habitat quality to climate change, as well as direct exploitation. This makes them an excellent indicator of anthropogenic changes in the environment.
The aim of the study was to determine long-term adaptation trends of AANAT copies in horseshoe populations in 11 geographic populations in China by sequencing and analyzing the selection pressures of AANAT genes.
Specific comments
Title:
The title of the article is adequate to the presented research results.
Abstract: could be supplemented with the most important statistical values
Introduction:
The authors of the article adequately presented the research problem, using the appropriate literature
Material and Methods
It is worth mentioning what was the individual number of each of the 11 populations. Whether the age and sex of the bats were determined.
Classic analytical procedures were used. The statistical analysis is unobjectionable.
Results:
The results are presented in the form of 7 graphs with descriptions and 5 tables. The results have been presented correctly, this chapter is very well drafted.
Discussion
The discussion chapter is too well described, it contains all the elements studied in the experiment. The cited literature is well selected.
Concluding remarks:
Why did the authors of the study not attempt to determine the level of melatons in the blood in particular populations and did not correlate the ANAT activity with the level of melatonin?
Data on the environment of individual populations should be supplemented, in particular with data on the diet of the species.
Author Response
We have uploaded the ”Cover Letter to Reviewer2“ to the attachment, please check.
